# Attribution-Based Confidence Metric For Deep Neural Networks

**Susmit Jha**
Computer Science Laboratory
SRI International

**Sunny Raj, Steven Lawrence Fernandes, Sumit Kumar Jha**
Computer Science Department
University of Central Florida, Orlando

**Somesh Jha**
University of Wisconsin-Madison
and Xaipient

**Brian Jalaian, Gunjan Verma, Ananthram Swami**
US Army Research Laboratory
Adelphi

## Abstract

We propose a novel confidence metric, namely, attribution-based confidence (ABC) for deep neural networks (DNNs). ABC metric characterizes whether the output of a DNN on an input can be trusted. DNNs are known to be brittle on inputs outside the training distribution and are, hence, susceptible to adversarial attacks. This fragility is compounded by a lack of effectively computable measures of model confidence that correlate well with the accuracy of DNNs. These factors have impeded the adoption of DNNs in high-assurance systems. The proposed ABC metric addresses these challenges. It does not require access to the training data, the use of ensembles, or the need to train a calibration model on a held-out validation set. Hence, the new metric is usable even when only a trained model is available for inference. We mathematically motivate the proposed metric and evaluate its effectiveness with two sets of experiments. First, we study the change in accuracy and the associated confidence over out-of-distribution inputs. Second, we consider several digital and physically realizable attacks such as FGSM, CW, DeepFool, PGD, and adversarial patch generation methods. The ABC metric is low on out-of-distribution data and adversarial examples, where the accuracy of the model is also low. These experiments demonstrate the effectiveness of the ABC metric towards creating more trustworthy and resilient DNNs.

## 1 Introduction

Deep neural network (DNN) models have been wildly successful in applications such as computer vision [1], natural language processing [2], and speech recognition [3, 4]. These models have reached human-level performance on several benchmarks [5, 3]. But the adoption of these models in safety-critical or high-assurance applications is inhibited due to two major concerns: their brittleness [6, 7] to adversarial attack methods or out-of-distribution inputs, and the lack of easily computable confidence measures that correlate well with the accuracy of the model. While improving the accuracy and robustness of DNNs has received significant attention, there is an urgent need to quantitatively characterize the limitations of these models and improve the transparency of their failure modes. This paper focuses on one such challenge: *defining a confidence metric on predictions of a DNN that closely reflects its accuracy*.

A confidence metric will be helpful in integrating DNN decision-making components in applications such as medical diagnosis [8] or autonomous systems [9] where such scores can be used to override the DNN decision using compositional architectures such as simplex [10]. A low confidence score can also be used to detect distribution shifts or adversarial attacks. A straightforward approach to computing such a score is to use the logit values prior to the softmax layer. But this raw confidence score is poorly calibrated [11, 12] and does not correlate well with prediction accuracy. The logits reflect high confidence even on wrong predictions over adversarial examples [13, 14, 6]. In fact, the improved accuracy in deep learning models over the last decade has been accompanied with worsening calibration of logit output to network accuracy [11] compared to earlier models [15]. It indicates overfitting in the negative log likelihood space even though DNNs avoid overfitting in the output accuracy [16]. This has motivated the development of confidence measures that use a held-out validation set for training a separate calibration model on the output of the DNNs [17, 18, 11]. But additional training or validation data may not be available with the trained machine learning (ML) model due to privacy, security or other practical constraints. Another class of confidence measures use sampling over model ensembles or training data to estimate conformance and data density [19–21]. In contrast, the focus of our work is to compute confidence for a DNN model on a given input without access to training data or the possibility of retraining a new model or model ensembles.

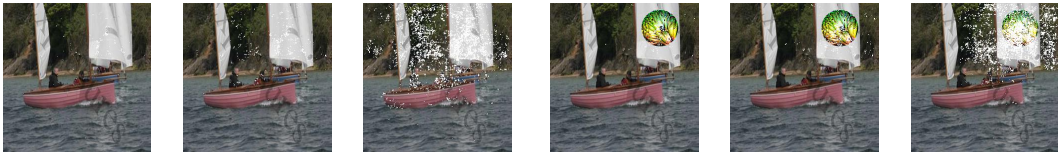

Figure 1: (From left to right) The original image with a label of yawl; masking its top $0.2\%$ attribution; masking its top $4\%$ of attribution; image with a banana adversarial patch [22]; masking its top $0.2\%$ attribution; (rightmost) masking its top $4\%$ of attribution. The classification result for the original image is robust and conforming in the neighborhood obtained by removing top positive attributions. The model predicts the original label (yawl) in the neighborhood. But the prediction changes from banana to yawl for many samples in the attribution neighborhood of the adversarial image. We observe a similar difference in conformance for the out-of-distribution examples and the adversarial examples generated by state-of-the-art digital attack methods such as DeepFool [23], PGD [24], FGSM [25] and Carlini-Wagner (CW) [26]. This observation motivates the use of conformance in the attribution-neighborhood as a confidence metric.

The confidence of a model on a given input can be measured by sampling the neighborhood of the input and observing whether the model's output changes or conforms to the original output. But accurately estimating the conformance by sampling in the neighborhood of the input becomes exponentially difficult with an increase in the dimension of the input [27]. We propose a novel approach which samples the high-dimensional neighborhood effectively using attributions provided by methods [28–30] developed recently to improve interpretability of DNNs. In particular, we adopt the use of Shapley values with roots in cooperative game theory for computing attribution. Approaches using Shapley values [30, 31] satisfy intuitively expected axiomatic properties over attributions [32]. Our attribution-based confidence metric is theoretically motivated by these axiomatic properties. The key idea is to use attributions for importance sampling in the neighborhood of an input. This importance sampling is more likely to select neighbors obtained by changing features with high attribution. The conformance of the machine learning model over these samples is computed as the fraction of neighbors for which the output of the model does not change. Figure 1 illustrates how an input that triggers incorrect responses in DNNs, such as an out-of-distribution sample or an adversarial example, does not conform in its neighborhood. Thus, the conformance of the model's prediction in the input's neighborhood, sampled using feature attributions, can be used as an effective measure of the confidence of the model on that input.

We make the following new contributions in the paper.

- We propose a novel attribution-based confidence (ABC) metric computed by importance sampling in the neighborhood of a high-dimensional input using relative feature attributions, and estimating conformance of the model. It does not require access to training data or additional calibration. We mathematically motivate the proposed ABC metric using axioms on Shapley values.

- We empirically evaluate the ABC metric over MNIST and ImageNet datasets using (a) out-of-distribution data, (b) adversarial inputs generated using digital attacks such as FGSM, PGD, CW and DeepFool, and (c) physically-realizable adversarial patches and LaVAN attacks.

## 2  Attribution-based Confidence (ABC) Metric

**Motivation:** The proposed ABC metric is motivated by Dual Process Theory [33, 34] and Kahneman's decomposition [35] of cognition into System 1, or the intuitive system, and System 2, or the deliberate system. The original DNN model represents the bottom-up System 1, and the ABC metric computation is a deliberative top-down System 2 that uses the attributions in System 1 to generate new samples in the neighborhood of the original input. Kilbertus et al. [36] argue that causal mechanisms are typically continuous but most learning problems such as classification are anti-causal. The lack of resilience to adversarial examples is hypothesized to be the result of learning in an anti-causal direction. Combining both the anti-causal System 1 DNN model and the attribution-driven System 2 that computes the ABC metric creates a relatively more resilient cognition model. ABC metric uses the attribution over features for the decision of a machine learning model on a given input to construct a generator that can sample the attribution-neighborhood of the input and observe the conformance of the model in this neighborhood. While learning is still in the anti-

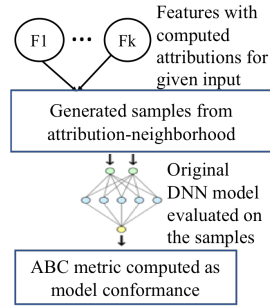

Figure 2: ABC complements the bottom-up inference of the original DNN model with the top-down sample generation and conformance estimation.

causal direction, ABC adds causal deliberative System 2 that reasons in the forward generative direction to evaluate the conformance of the model.

**Computational challenge:** The computation of ABC metric of an ML model on an input requires accurately determining conformance by sampling in the neighborhood of high-dimensional inputs. This can be addressed by sampling over lower-dimensional intermediate or output layers of a DNN [19, 17], or relying on topological and manifold-based data analysis [20]. But these methods require training data that may not always be available at inference time. ABC addresses this challenge by biasing our sampling using the quantitative attribution obtained via Shapley values [30]. Deep learning models demonstrate a concentration of features, that is, few features have

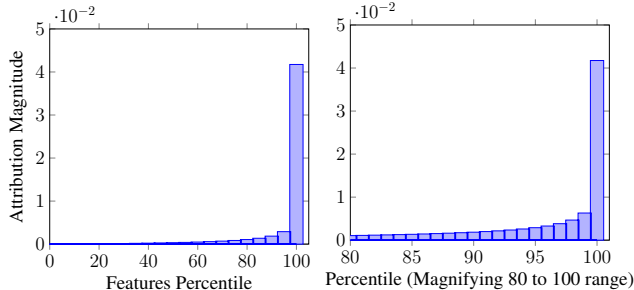

Figure 3: Attributions concentrate over few features in ImageNet.

relatively very high attributions for any decision. Figure 3 illustrates feature concentration for ImageNet where the attribution is computed via Integrated Gradients [30]. Sampling over low attribution features will likely lead to no change in the label. Low attribution indicates that model is *equivariant* along these features. By focusing on high attribution features during sampling, our method can efficiently sample even high-dimensional input spaces to obtain a conservative estimate of the confidence score.

Given an input $\mathbf{x}$ for a model $\mathcal{F}$ where $\mathcal{F}_i$ denotes the $i$-th logit output of the model, we can compute attribution of feature $\mathbf{x}_j$ of $\mathbf{x}$ for label $i$ as $\mathcal{A}_j^i(\mathbf{x})$. We can then compute ABC metric in two steps:

- Sample the neighborhood: Select feature $\mathbf{x}_j$ with probability $\frac{|\mathcal{A}_j^i(\mathbf{x})/\mathbf{x}_j|}{\sum_j |\mathcal{A}_j^i(\mathbf{x})/\mathbf{x}_j|}$ and change it to flip the label away from $i$, that is, change the decision of the model (for example, by changing the feature's value to a baseline used in computing the attribution).

- Measure conformance: Report the fraction of samples in the neighborhood for which the decision of the model does not change, that is, conforms to the original decision, as the conservatively estimated confidence measure.

ABC uses feature attributions for dimensionality reduction followed by importance sampling in the reduced-dimensional neighborhood of the input to estimate DNN model's conformance. But unlike typical principal component analysis techniques that search for globally important features, we identify features that are locally relevant for the given input. This enables our approach to conservatively approximate conformance measure of a model in even high-dimensional input's neighborhood and, thus, efficiently compute the ABC confidence metric of the model on the input.

# 3   ABC Algorithm

In this section, we theoretically motivate the proposed ABC metric and present an algorithm for its computation. Attribution methods using Shapley values often employ the notion of a baseline input $\mathbf{x}^b$; for example, the all dark image can be the baseline for images. The baseline can also be a set of random inputs where attribution is computed as an expected value. Let the attribution for the $j$-th feature and output label $i$ be $\mathcal{A}_j^i(\mathbf{x})$. The attribution for the $j$-th input feature depends on the complete input $x$ and not just $x_j$. The treatment for each logit is similar, and so, we drop the logit/class and denote the network output simply as $\mathcal{F}(\cdot)$ and attribution as $\mathcal{A}_j(\mathbf{x})$. For simplicity, we use the baseline input $\mathbf{x}^b = 0$ for computing attributions. We make the following two assumptions on the DNN model and the attributions, which reflect the fact that the model is well-trained and the attribution method is well-founded:

- The attribution is dominated by the linear term. This is also an assumption made by attribution methods based on Shapley values such as Integrated Gradient [30] which define attribution as the path integral of the gradients of the DNN output with respect to that feature along the path from the baseline $\mathbf{x}^b$ to the input $\mathbf{x}$, that is,

$$\mathcal{A}_j^i(\mathbf{x}) = (\mathbf{x}_j - \mathbf{x}_j^b) \times \int_{\alpha=0}^{1} \partial_j \mathcal{F}^i(\mathbf{x}^b + \alpha(\mathbf{x} - \mathbf{x}^b))d\alpha \tag{1}$$

  where the gradient of $i$-th logit output of the model along the $j$-th feature is denoted by $\partial_j \mathcal{F}^i(\cdot)$.

- Attributions are complete i.e. the following is true for any input $\mathbf{x}$ and the baseline input $\mathbf{x}^b$:

$$\mathcal{F}(\mathbf{x}) - \mathcal{F}(\mathbf{x}^b) = \sum_{k=1}^{n} \mathcal{A}_k(\mathbf{x}) \text{ where } \mathbf{x} \text{ has } n \text{ features.} \tag{2}$$

  Shapley value methods such as Integrated Gradient and DeepShap [30, 31] satisfy this axiom too.

We first establish a relationship between attributions and the sensitivity of the model's output to change in an input feature. This is useful in attribution-based dimensionality reduction of a high-dimensional $\mathbf{x}$ and defining its attribution-neighborhood.

**Theorem 1.** *The sensitivity of the output $\mathcal{F}(\mathbf{x})$ with respect to an input feature $\mathbf{x}_j$ in the neighborhood of $\mathbf{x}$ is approximately the ratio of the attribution $\mathcal{A}_j(\mathbf{x})$ to the value of that feature $\mathbf{x}_j$, that is, $\frac{\mathcal{A}_j(\mathbf{x})}{\mathbf{x}_j}$.*

*Proof.* Given an input $\mathbf{x}$ and its neighbor $\mathbf{x}' = \mathbf{x} + \delta\mathbf{x}$, we can use Taylor series expansion to express the output $\mathcal{F}(\mathbf{x}')$ as:

$$\mathcal{F}(\mathbf{x}') = \mathcal{F}(\mathbf{x}) + \sum_{k=1}^{n} \left( \frac{\partial \mathcal{F}(\mathbf{x})}{\partial \mathbf{x}_k} \delta\mathbf{x}_k \right) + \max_{k=1,\dots,n} O(\delta\mathbf{x}_k^2) . \tag{3}$$

Assuming the completeness of attribution over the features of the input $\mathbf{x}'$, we obtain the following:

$$\mathcal{F}(\mathbf{x}') - \mathcal{F}(\mathbf{x}^b) = \sum_{k=1}^{n} \mathcal{A}_k(\mathbf{x}') . \tag{4}$$

Subtracting Equation 2 from 4, we can eliminate the baseline input $\mathbf{x}^b$ and use Taylor series to obtain:

$$\begin{aligned}
\mathcal{F}(\mathbf{x}') - \mathcal{F}(\mathbf{x}) &= \sum_{k=1}^{n} \left( \mathcal{A}_k(\mathbf{x}') - \mathcal{A}_k(\mathbf{x}) \right) \\
&= \sum_{k=1}^{n} \left( \frac{\partial \mathcal{A}_k(\mathbf{x})}{\partial \mathbf{x}_k} \delta\mathbf{x}_k \right) + \max_{k=1,\dots,n} O(\delta\mathbf{x}_k^2)
\end{aligned} \tag{5}$$

While computing conformance, we sample only neighbors of the input and so we can drop the higher order terms in both Equations 5 and 3. These equations hold for all neighbors including those which differ in only one of the features $\mathbf{x}_j$ and so, we can conclude that the sensitivity of the model with respect to a feature $\mathbf{x}_j$ is $\frac{\partial \mathcal{A}_j(\mathbf{x})}{\partial \mathbf{x}_j}$.

Taking the derivative of Equation 1 on both sides with respect to feature $\mathbf{x}_j$ and ignoring the non-linear attribution terms, we obtain the following:

$$\frac{\partial \mathcal{A}_j(\mathbf{x})}{\partial \mathbf{x}_j} = \int_{\alpha=0}^{1} \frac{\partial \mathcal{F}(\mathbf{x}^{\mathbf{b}} + \alpha(\mathbf{x} - \mathbf{x}^b))}{\partial \mathbf{x}_j} d\alpha + \mathbf{x}_j \frac{\partial}{\partial \mathbf{x}_j} \left( \int_{\alpha=0}^{1} \frac{\partial \mathcal{F}(\mathbf{x}^{\mathbf{b}} + \alpha(\mathbf{x} - \mathbf{x}^b))}{\partial \mathbf{x}_j} d\alpha \right)$$

$$= \int_{\alpha=0}^{1} \frac{\partial \mathcal{F}(\mathbf{x}^{\mathbf{b}} + \alpha(\mathbf{x} - \mathbf{x}^b))}{\partial \mathbf{x}_j} d\alpha + \mathbf{x}_j \left( \int_{\alpha=0}^{1} \frac{\partial^2 \mathcal{F}(\mathbf{x}^{\mathbf{b}} + \alpha(\mathbf{x} - \mathbf{x}^b))}{\partial \mathbf{x}_j^2} d\alpha \right)$$

$$\approx \int_{\alpha=0}^{1} \frac{\partial \mathcal{F}(\mathbf{x}^{\mathbf{b}} + \alpha(\mathbf{x} - \mathbf{x}^b))}{\partial \mathbf{x}_j} d\alpha = \frac{\mathcal{A}_j(\mathbf{x})}{\mathbf{x}_j} \text{ from Eqn. 1 with baseline feature } \mathbf{x}_j^b = 0.$$

(6)

Thus, we have shown that the sensitivity of the model's output with respect to an input feature in the neighborhood of the input is approximately given by the ratio of the attribution for that input feature to the value of that feature. Note that $\frac{\mathcal{A}_j(\mathbf{x})}{\mathbf{x}_j}$ does not vanish even when the traditional sensitivity given by $\frac{\partial \mathcal{F}(\mathbf{x})}{\partial \mathbf{x}_j}$ vanishes exploiting the non-saturating nature of Shapley value attributions. $\square$

The model is almost equivariant with respect to features with low $|\mathcal{A}_j(\mathbf{x})/\mathbf{x}_j|$ and, thus, the attribution-neighborhood is constructed by mutating features with high $|\mathcal{A}_j(\mathbf{x})/\mathbf{x}_j|$. The overall algorithm for computing ABC metric of a DNN model on an input is as follows.

---

**Algorithm 1** Evaluate ABC confidence metric $c(\mathcal{F}, \mathbf{x})$ of machine learning model $\mathcal{F}$ on input $\mathbf{x}$

---

**Input:** Model $\mathcal{F}$, Input $\mathbf{x}$ with features $\mathbf{x}_1, \mathbf{x}_2, \ldots \mathbf{x}_n$, Sample size $S$
**Output:** ABC metric $c(\mathcal{F}, \mathbf{x})$
1: $\mathcal{A}_1, \ldots \mathcal{A}_n \leftarrow$ Attributions of features $\mathbf{x}_1, \mathbf{x}_2, \ldots \mathbf{x}_n$ from input $\mathbf{x}$
2: $i \leftarrow \mathcal{F}(\mathbf{x})$ {Obtain model prediction}
3: **for** $j = 1$ to $n$ **do**
4: $\quad P(\mathbf{x}_j) \leftarrow \frac{|\mathcal{A}_j/\mathbf{x}_j|}{\sum_{k=1}^{n} |\mathcal{A}_k/\mathbf{x}_k|}$
5: **end for**
6: Generate $S$ samples by mutating feature $\mathbf{x}_j$ of input $\mathbf{x}$ to baseline $\mathbf{x}_j^b$ with probability $P(\mathbf{x}_j)$
7: Obtain the output of the model on the $S$ samples.
8: $c(\mathcal{F}, \mathbf{x}) \leftarrow S_{conform}/S$ where model's output on $S_{conform}$ samples is $i$
9: **return** $c(\mathcal{F}, \mathbf{x})$ as confidence metric (ABC) of prediction by the model $\mathcal{F}$ on the input $\mathbf{x}$

---

## 4 Related Work: Confidence, Robustness and Interpretability

Our proposed confidence metric (ABC) is closely related to the literature on confidence metrics, attribution methods and techniques for adversarial attacks and defenses for DNNs.

**Confidence metrics:** The need for confidence metric to reflect the uncertainty in the output of machine learning models was recognized very early in the literature [18, 17]. The high accuracy but brittleness of deep learning models has revived interest in defining confidence metrics that reflect the accuracy of the model. DNNs are not well-calibrated [11, 19, 20] and so, the straightforward use of logit layer before softmax as a confidence measure is not reliable. Several post-processing based confidence metrics have been proposed in the literature which can be grouped into three classes:

- Calibration models trained using held-out validation set: Platt [18] proposed a parametric approach where the logits are used as features to learn a calibration model from a held-out validation set. An example calibration model is $q_i = \max_k \sigma(W z_i + b)^k$ where $z_i$ are the logits, $\sigma$ is the standard sigmoid function, $k$ denotes a class, $(\cdot)^k$ denotes the $k$-th element of a vector, and $W, b$ are parameters [15]. Temperature scaling is a special case of Platt scaling with a single parameter. Nonparametric learning of calibration models from held-out validation data has been also proposed in the form of histogram binning [37], isotonic regression [38] and Bayesian binning [39].
- Model ensemble approaches: Lakshminarayanan et al. [40] use ensembles of networks to obtain uncertainty estimates. Bayesian neural networks [17, 41] return a probability distribution over outputs as an alternative way to represent model uncertainty. Sampling models by dropping nodes in a DNN has been shown to estimate probability distribution over all models [42].

- Training-set-based uncertainty estimation: Jiang et al. [20] compute the trust score on deeper layers of a DNN than input to avoid the high-dimensionality of inputs. They propose a trust score that measures the conformance between the classifer and a modified nearest-neighbor classifier on the testing example. Papernot and McDaniel [19] use $k$-nearest neighbors regression using training set on the intermediate representations of the network which showed enhanced robustness to adversarial attacks and leads to better calibrated uncertainty estimates.

In contrast to these approaches, ABC metric only needs the trained model at inference time, and does not require training data, separate held-out validation data or training a model ensemble.

**DNN robustness, adversarial inputs and defense:** Szegedy et al. [6] used L-BFGS method to generate adversarial examples. Goodfellow [25] proposed a fast gradient sign method (FGSM) to generate faster adversarial images as compared to the L-BFGS method; this method performed only a one-step gradient update at each pixel along the direction of the gradient's sign. Rozsa et al. [43] replaced the sign of the gradients with raw gradients. Dong et al. [44] applied momentum to FGSM, and Kurakin et al. [45] further extended FGSM to a targeted attack. A number of adversarial example generation techniques [23, 24, 26] are now available in tools such as Cleverhans [7] which we use in our experiments. In addition to digital attacks, we also study how the proposed ABC metric measures uncertainty on physically realizable attacks in form of patches or stickers [22, 46]. Further, the adversarial examples have been shown to transfer [47, 48] across models making them agnostic to availability of model parameters and effective against even ensemble approaches. Efforts over the last few years to defend against adversarial attacks have met with limited success. While approaches such as logit pairing [49], defensive distillation [50], manifold-based defense [51, 52], and adversarial training methods that exploit knowledge of the specific attack [53], have shown effectiveness against particular attack methods, more principled techniques such as robust optimization [54, 55, 24] and formal methods [56] are limited to perturbations with bounded $L_p$ norm. Schott et al. [57] present an insightful study of the state of the art on attacks and defenses.

Our approach to addressing the challenge of DNN's susceptibility to adversarial examples using ABC metric differs from the majority of prior work in that it measures confidence of a model's prediction to characterize the credibility of a DNN on a given input instead of attempting to classify all legitimate and malicious inputs correctly or make particular adversarial strategies fail.

**Model interpretability and attribution methods:** A number of explanation techniques [31, 30, 58–61] have been recently proposed in the literature that either find a complete logical explanation or just the relevant features or assign quantitative importance (attributions) to input features for a given model decision. Many of these methods are based on the gradient of the predictor function with respect to the input [28–30]. Different attribution methods are compared in Adebayo et al. [62]. The sensitivity of these attributions to perturbations in the input are studied in Ghorbani et al. [63], and indicate the potential of adversarial attacks on the proposed and other existing confidence metrics. But attacking the model and its confidence measure is more difficult than just attacking the model. This motivates the use of ABC metric to measure confidence. The brittleness of learning has been related to anti-causal direction of learning [36]. We observe that the wrong decisions for out-of-distribution and adversarial inputs often hinge on a relatively small concentrated set of high attribution features, and thus, mutating these features and generating samples in the neighborhood of the original input is an effective top-down inference that is robust to adversarial attacks. In recent work [64], we have demonstrated the use of attributions for detecting adversarial examples.

## 5   Experimental Evaluation

We evaluate the attribution-based confidence metric on out-of-distribution data and adversarial attacks. All experiments were conducted on a 8 core Intel® Core™ i9-9900K 3.60GHz CPU with NVIDIA Titan RTX Graphics and 32 GB RAM.

**Out-of-distribution data: MNIST [65] with rotation and background, notMNIST [66] and FashionMNIST [67].** We compare the predicted accuracy vs confidence below to evaluate how well the attribution-based confidence metric reflects the reduced accuracy on the rotated MNIST dataset with a background image [68]. The dataset has MNIST images randomly rotated by $0$ to $2\pi$, and with a randomly selected black/white background image. The accuracy and confidence of the model drops with increase in rotation angle (from $0$ to $50$ degrees) and decrease in accuracy as illustrated in Figure 4. The three examples show how the ABC metric reflects the confusability of inputs.

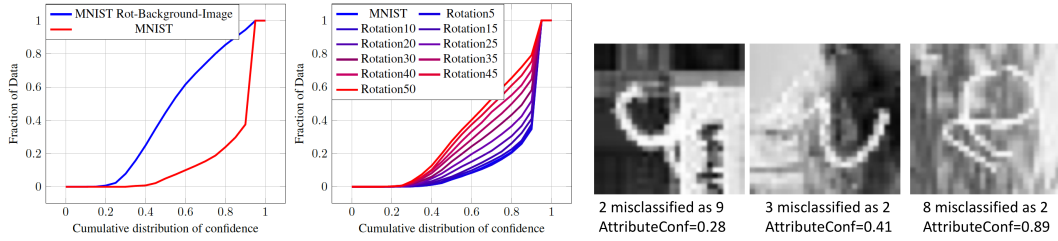

Figure 4: ABC for rotated-background-MNIST and rotated-MNIST at different angles. Selected examples from rotated-background-MNIST with confidence showing quantitative analysis of ABC metric.

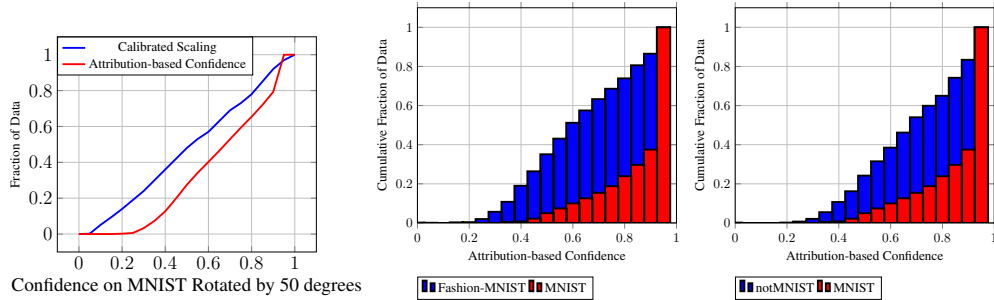

Figure 5: Comparison with a calibrated scaling model. Cumulative data fraction vs. ABC for FashionMNIST and nonMNIST compared with MNIST. Only $19\%$ of the FashionMNIST dataset and $26\%$ of the notMNIST dataset have a confidence higher than $0.85$ while $70\%$ of MNIST dataset had confidence higher than $0.85$.

We compare the ABC metric with a trained calibrated scaling model and also evaluate it on out-of-distribution FashionMNIST and notMNIST datasets in Figure 5.

**Adversarial FGSM and PGD attacks on MNIST.** Figure 6 illustrates how the decreased ABC metric reflects the decrease in accuracy on adversarial examples. The accuracy with PGD attack drops close to zero and so, we only plot the fraction of data with different confidence levels.

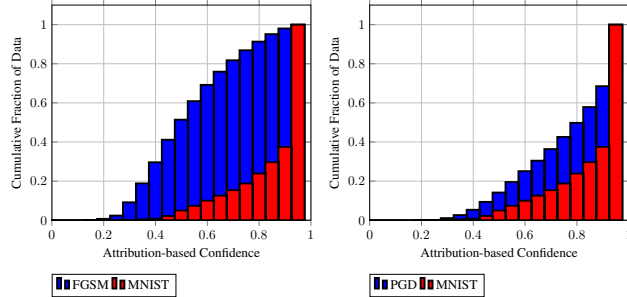

Figure 6: ABC metric for FGSM and PGD attacks on MNIST.

**Comparison of attribution methods.** We also compare different attribution methods in Figure 7.

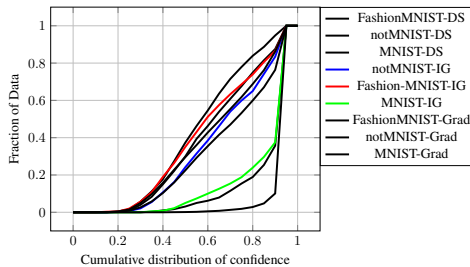

Figure 7: Comparing ABC metric using different attribution methods: Gradients (Grad), Integrated Gradient (IG), and DeepSHAP (DS). For out of distribution examples (FashionMNIST and notMNIST), results with DeepShap are slightly better than IG (which is better than Gradient).

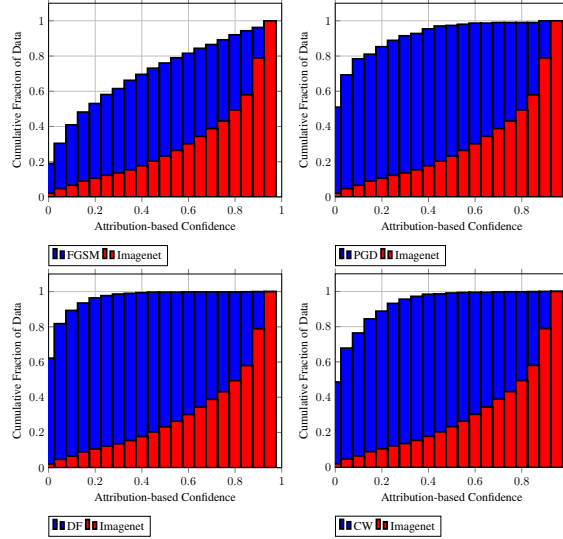

Figure 8: ABC metric for FGSM, PGD, DeepFool and CW attacks on ImageNet.

**Digital Adversarial attacks on ImageNet [69]: FGSM, PGD, CW and DeepFool.**    Figure 8 illustrates how the ABC metric reflects the decrease in accuracy under adversarial attack.

**Physically realizable adversarial patch attacks on ImageNet.**    We apply physically realizable adversarial patch [22] and LaVAN [46] attacks on 1000 images from ImageNet. For adversarial patch attack, we used a patch size of 25% for two patch types: banana and toaster. For LaVAN, we used baseball patch of size $50 \times 50$ pixels. Figure 9 illustrates how the ABC metric is low for most of the adversarial examples reflecting the decrease in the accuracy of the model.

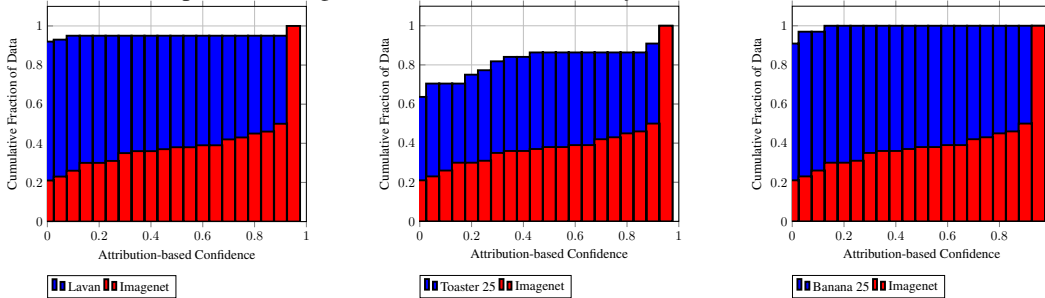

Figure 9: Cumulative data fraction vs. ABC metric for ImageNet and adversarial patch attacks.

## 6    Conclusion

We employ an attribution-driven sampling of the neighborhood of a given input and measure the conformance of the model's predictions to compute the attribution-based confidence (ABC) metric for DNN prediction on this input. While directly sampling the neighborhood of a high-dimensional input is challenging, our approach uses attribution-based dimensionality reduction for finding locally relevant features in the vicinity of the input, which enables effective sampling. We theoretically motivate the proposed ABC metric from the axioms of Shapley values, and experimentally evaluate its utility over out-of-distribution data and adversarial examples.

**Acknowledgement:** The authors acknowledge support from the U.S. Army Research Laboratory Cooperative Research Agreement W911NF-17-2-0196, DARPA Assured Autonomy under contract FA8750-19-C-0089, U.S. National Science Foundation(NSF) grants #1422257, #1740079, #1750009, #1822976, #1836978, #1804648, ARO grant W911NF-17-1-0405, Royal Bank of Canada, Cyber Florida, U.S. Air Force Young Investigator award, and FA9550-18-1-0166. The views, opinions and/or findings expressed are those of the author(s) and should not be interpreted as representing the official views or policies of the Department of Defense or the U.S. Government.

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
