[Reviews · NeurIPS 2019]

Reviewer 1



In this paper the authors propose a confidence estimation method which relies feature perturbation. However this method avoids perturbation in high dimensional feature space by selecting a subset of features through an attribution method. They provide theoretical justification for using input gradients as a substitute for attribution. The paper provides a rich literature survey which can aid readers newly approaching this area of research. Further they span their survey over different classes of approaches which provides a better understanding to the readers. Empirical evaluations are provided for out-of-distribution detection, adversarial example detection. Further they also provided evaluations for both digital and physically realizable adversarial patch attacks.

Reviewer 2



Overall Comments This paper is reasonably well motivated and provide justifications for the key use of integrated gradients as part of the computing the confidence score. The paper also presents several empirical demonstrations of the algorithm. The key motivation is that one might want to compute calibration scores without retraining like is typical for isotonic regression and platt scaling. Originality I am not aware of work using integrated gradients for computing calibration scores. However, the literature on interpretability and uncertainty representation is vast. In this regard though, the paper does a pretty comprehensive job covering several recent work. Quality The work is well motivated and demonstrates that this type of technique is needed. In addition several of the experiments do indeed show that the work captures some sense of uncertainty that is typically reflected in modern neural networks. One piece that is currently missing from this work in my opinion is a clear and solid proof of concept; I'll expound on this later in this review. Clarity This paper is reasonably well written and I was able to follow several of the clarifications behind this work. A few suggestions, the abstract as it currently stands is quite dense; the authors could easily cut out a few sentences and still retain the overall story. This is a matter of style so feel free to disregard, but the first four sentences of the abstract could be easily cut without any loss in meaning. This is the same for large parts of the introduction. The second paragraph of the introduction could be broken into two and condensed more succinctly. Sentence starting with "Robust machine learning models ..." on line 97 seems to me to be a matter of opinion. One could easily build a robust model, (not sure what definition of robust you have here), that still places a significant amount of attribution on a lot of inputs dimensions. I'll return to this point later on. I like section 3; it is clear and a good mathematical motivation of your work. Significance The motivation of this paper is significant, and as far as I can tell, there are not a lot of works that have proposed this metric before. Some Questions and Points to address - Need for a proof or concept or demo. I know reviewers are often accused of asking for extra experiments during rebuttals, which is not often realistic; however, here is one that would better convince me of the utility of this work. Train, say a logistic regression classifier on a dataset where you can reasonably control how well calibrated this classifier is. Now get the 'ground truth' classification with platt scaling or one of the baseline methods that you mention, finally compare this 'ground truth' classification to ones derived with Algorithm 1. This will greatly nail your point home. - Sparse of Integrated Gradients Attribution. One key motivation in this paper for using integrated gradients is that it is not efficient to sample in the neighborhood of high-dimensional inputs, so one can use the attribution to weight the kinds of inputs that one samples. This is a fine motivation; however, it is not clear to me that integrated gradients attributions should be sparse. For things like MNIST, and Fashion MNIST where the background is black, the attributions are typically sparse, however, in looking at examples of integrated gradients attributions on Imagenet, these don't look sparse, so it is not clear how much the IG attributions will actually help. Can the authors comment on this? - Too much focus on adversarial examples in demonstration. This paper spends a lot of time showing that algorithm 1 produces low confidence on adversarial examples. Sure, adversarial examples are extremely popular topic these days, however, calibration and uncertainty by themselves are very important topics, and I would encourage the authors to look beyond this specific application. I stress this because, an adversary that takes into account the fact that you are using an IG attribution map can fool your method to produce high confidence scores on adversarial examples. To send this point home, the methods in citation 57 are exactly what one would need to do this. Update After reading the author rebuttable, I am planning to stick with an accept for this work. The authors have provided baseline comparisons and answers more clarifying questions. I think this is a valuable contribution that could spur future work in this direction.

Reviewer 3



Update after author response: - I thank the authors for their response and for the work they put in. I see now that the derivation for eqn 7 is clear; I think I forgot to include the detail that xb was 0 when I tried to replicate the derivation, for which I apologize. However, it seems that to get from the second line of eqn 7 to the third line of eqn 7, the term with the second derivative is dropped. I am not sure whether this is justified, and it is probably the reason for the counter-intuitive conclusion of that theorem. But more importantly, I don't feel Theorem 1 is even necessary for the method as a whole to make sense; I think people will accept the statement that features that have low importance are not likely to change the output when perturbed, assuming that the importance scoring method is good, even if there are edge cases where this may not be true. - I appreciate that the authors included a comparison with Platt scaling. From my understanding of Figure 2, it seems that Platt Scaling also assigns lower confidence to out-of-distribution examples, as reported in the prior literature. It would be interesting if the authors could combine the confidence rating from Platt Scaling with the confidence rating from their proposed method to develop and even more effective algorithm than Platt Scaling alone. I know the authors said that the advantage over Platt Scaling is that they don't require access to a held-out validation set for calibration, but in practice people do tend to have a held-out validation set, so I am not sure the need for a held-out validation set is much of a barrier. - I also appreciate that the authors included a quantitative evaluation in Figure 3. It would have been ideal if the authors had reported the results from Platt scaling in this figure too. - It is interesting to see that DeepSHAP outperforms Integrated Gradients and the authors should definitely include this in the paper. Since it is not clear to me that the method does better than Platt Scaling, and I think in most practical situations people do have access to a held-out validation set for calibration, I am leaving my rating at "marginally above acceptance threshold"; the paper has value in terms of the originality of the attribution method proposed, but appears to need some further work before it can become a practical tool that people will use. If the authors could find a way to combine their proposed confidence metric with baselines from the literature to find something that works better than the baselines from the literature, I think that would be what is needed to encourage adoption. --- I tend to vote in favor of this submission because I think the core idea, that of rating the confidence of examples according to how easily the prediction changes when the examples are perturbed, is a potentially very promising approach that has not been explored enough in the literature. The major drawbacks are (1) Theorem 1 seems suspicious given the ability of Integrated Gradients to highlight the importance of saturated inputs where the partial derivative is locally zero, (2) the choice of all all-zeros baseline may not be a good idea, and (3) the empirical benchmarking could be improved. Originality As mentioned, I think the core idea of rating the confidence of examples according to how easily the prediction changes when the example is perturbed is clever. Although this approach may have been used for other types of ML models, I haven't yet seen it adapted to the context of neural networks. The idea of using some form of importance scoring to prioritize the perturbations also makes a lot of sense. Quality - I feel that the empirical benchmarking could be improved. Minimally, I feel that the authors should compare to a baseline that uses calibrated predicted probabilities. The authors do discuss the baseline of calibrated probabilities in the related works section and argue that a downside of this approach is that it requires access to a held-out validation set for calibration. However, modern neural networks are almost always trained with a held-out validation set for early stopping and hyperparameter search, so there are not many cases where a held-out validation set is unavailable. In the caption of Figure 4, they said "The softmax confidence (green) is always high and does not reflect accuracy", but it was unclear whether this softmax confidence was calibrated. - I would have also appreciated seeing more quantitative evaluations. For example, if examples were ranked according to the confidence score, what would be the auROC for identifying misclassified examples? Alternatively, they could quantify by how much the performance of the classifier improves if the least-confident examples are abstained on. Quantitative metrics such as these would make it easier to compare different methods. By simply viewing the distribution plots, I cannot clearly tell, for example, whether the confidence estimates from the Bayesian model are better or worse than the confidence estimates obtained by the proposed method (though I understand that the proposed method has several advantages over building a Bayesian model). - I am concerned about the choice of an all-zero baseline for Integrated Gradients. The choice of baseline is a very importance consideration for reference-based interpretation methods (see https://github.com/kundajelab/deeplift#what-should-i-use-as-my-reference), and using an all-zeros baseline may not be a good idea, because it will make it impossible to highlight pixels that are black (as those pixels will be equal to their baseline value). It may be better to sample several different reference values and average the results over them, as is done in the DeepSHAP implementation. An approach like DeepSHAP would also have the advantage of being computationally faster than Integrated Gradients. Clarity Theorem 1 also seems suspicious because, as mentioned earlier, one of the main selling points of Integrated Gradients is the ability to assign importance to features that have saturated in terms of their partial derivatives. In other words, Integrated Gradients is designed such that it can assign high importance to features for which the partial derivative is zero. This seems quite contrary to the statement of Theorem 1. Significance I think the core idea of this paper - that of assigning confidence for a deep learning model based on how much the predictions change when an example is perturbed - has the potential to catch on. At its core, this method is like the nearest-neighbors-based density approaches - however it does not require access to the training set at prediction time, which is a major advantage. The idea of sampling perturbations based on some form of an attribution also has the potential to catch on, in my opinion. I am less sold on the use of Integrated Gradients with a zero baseline, and I think the empirical evaluation could be improved, but to me the core idea makes a lot of sense even if the evaluation is a bit lacking.

[Author Response · NeurIPS 2019]

We thank the reviewers for encouraging and insightful comments. We clarify the major points below:

**Reviewer 3, Comment 3.1. Evaluating the confidence met-**
**ric over more confusable out-of-distribution examples.**
A model trained on MNIST dataset was run on a more *con-*
*fusing* dataset, `MNIST-rot-back-image`[1]. The accuracy and
confidence of the model drops for the confusing dataset (see
Figure 1 (Top Left)). Results in Fig. 1 (Top Right) show that
the attribution-based confidence drops with increase in rotation
angle (from 0 to 50 degrees) and decrease in accuracy.
**Reviewer 3, Comment 3.2. Interpretability and qualitative**
**analysis of the confidence metric.** Figure 1 (Bottom) illus-
trates how confidence computed by attribution on examples
from `MNIST-rot-back-image` reflects the perceived ambigu-
ity and confusability of inputs.

**Reviewer 4, Comment 4.1. Demo against Platt scal-**
**ing/Calibrated predictor baseline.** The comparison of
attribution-based confidence metric with calibrated Platt scal-
ing model is shown in Figure 2 (Left).
**Reviewer 4, Comment 4.2. The sparseness of IG attribu-**
**tion maps.** We present the distribution of attributions for
ImageNet in Figure 2 (Right). As anticipated, attributions
concentrate over a small number of high-attribution features.

2 misclassified as 9
AttributeConf=0.28

3 misclassified as 2
AttributeConf=0.41

8 misclassified as 2
AttributeConf=0.89

Figure 1: Confidence computed on new dataset `MNIST-rot-back-image` (Top Left) and rotated MNIST (Top Right) at different angles. Selected examples from `MNIST-rot-back-image` (Bottom).

**Reviewer 4, Comment 4.3. Less fo-**
**cus on adversarial examples.** We will
supplement our earlier reported results
on out-of-distribution examples with
additional new results (such as those
presented here) to balance focus away
from adversarial examples.

Figure 2: (Left 2) Comparison with calibrated Platt (temperature) scaling
model. (Right 2) Concentration of Attributions over few features for ImageNet.

30 **Reviewer 5, Comment 5.1. Theorem 1: How**
31 **the first line of equation was arrived?; Rec-**
32 **oncile with IG assigning high importance to**
33 **saturating inputs.** The first line of equation is
34 derived by using the product rule for differen-
35 tiation. that is, $\frac{d(fg)}{dx} = f.\frac{dg}{dx} + \frac{df}{dx}.g$. The IG
36 attribution is $\mathcal{A}_j^i(\mathbf{x}) = (\mathbf{x}_j - \mathbf{x}_j^b) \times \int_{\alpha=0}^1 \partial_j \mathcal{F}^i(\mathbf{x}^b +$
37 $\alpha(\mathbf{x} - \mathbf{x}^b))d\alpha$. By differentiating w.r.t $x_j$ using
38 product rule, we get $\int_{\alpha=0}^1 \frac{\partial \mathcal{F}^i(\mathbf{x}^b + \alpha(\mathbf{x} - \mathbf{x}^b))}{\partial \mathbf{x}_j} d\alpha +$
39 $(\mathbf{x}_j - \mathbf{x}_b)\frac{\partial}{\partial \mathbf{x}_j}\left(\int_{\alpha=0}^1 \frac{\partial \mathcal{F}^i(\mathbf{x}^b + \alpha(\mathbf{x} - \mathbf{x}^b))}{\partial \mathbf{x}_j} d\alpha\right)$ where
40 $\mathbf{x}_b$ is set to 0. The IG attribution is non-zero
41 even when the partial derivative is zero (en-
42 abling better measurement for saturating fea-
43 tures) but IG attribution also saturates, albeit to
44 a non-zero value. Consequently, we avoid the

Figure 3: (Left) Accuracy improvement after abstention on `MNIST-rot-back-image`. (Right) Comparing different implementations of attribution-based confidence using Gradients (Grad), Integrated Gradient (IG), and DeepSHAP (DS). For out of distribution examples (FashionMNIST and notMNIST), results with DeepShap are slightly better than IG (which is better than Gradient).

45 effect of saturation while using the change in attribution for importance sampling by approximating the rate of change
46 of attribution in Eqn. 7 as a linear variation over the change in features. We will include this discussion and associated
47 intuition with an illustrative example.
48 **Reviewer 5, Comment 5.2. Platt scaling comparison.** Please see Figure 2 (Left).
49 **Reviewer 5, Comment 5.3. Improvement after abstention.** Please see Figure 3 (Left).
50 **Reviewer 5, Comment 5.4-5.5. Using DeepShap and Gradients.** Please see Figure 3 (Right) for results on MNIST,
51 notMNIST, and FashionMNIST. More detailed evaluation with DeepShap/Gradients will be included in the full paper.

## Footnotes

[1]Public dataset from U.Montreal (link omitted because response must not have external links). Dataset has MNIST images randomly rotated by 0 to $2\pi$, and with a randomly selected black/white background image. See examples in Fig. 1 (Bottom).


[Meta-Review · NeurIPS 2019]

This paper is well-motivated, written clearly, and provides theoretical and empirical evidence for the utility of integrated gradients for computing confidence scores of neural networks. The ideas presented are novel and are backed up quite well with theory and experiments. Few suggestions for improvement in the final version of the paper: 1. simple demo against Platt scaling 2. clarification of the sparseness of IG attribution maps 3. a more detailed qualitative error analysis of confidence metric All in all, this is a good contribution and I recommend its acceptance.